# Learning-to-Rank Meets Language: Boosting Language-Driven Ordering Alignment for Ordinal Classification

**Rui Wang[1], Peipei Li[1][\*], Huaibo Huang[2], Chunshui Cao[3], Ran He[2], Zhaofeng He[1]**
[1]Beijing University of Posts and Telecommunications
[2]CRIPAC&MAIS, Institute of Automation, Chinese Academy of Sciences
[3]WATRIX.AI
{wr_bupt, lipeipei, zhaofenghe}@bupt.edu.cn
huaibo.huang@cripac.ia.ac.cn, chunshui.cao@watrix.ai, rhe@nlpr.ia.ac.cn

## Abstract

We present a novel language-driven ordering alignment method for ordinal classification. The labels in ordinal classification contain additional ordering relations, making them prone to overfitting when relying solely on training data. Recent developments in pre-trained vision-language models inspire us to leverage the rich ordinal priors in human language by converting the original task into a vision-language alignment task. Consequently, we propose L2RCLIP, which fully utilizes the language priors from two perspectives. First, we introduce a complementary prompt tuning technique called RankFormer, designed to enhance the ordering relation of original rank prompts. It employs token-level attention with residual-style prompt blending in the word embedding space. Second, to further incorporate language priors, we revisit the approximate bound optimization of vanilla cross-entropy loss and restructure it within the cross-modal embedding space. Consequently, we propose a cross-modal ordinal pairwise loss to refine the CLIP feature space, where texts and images maintain both semantic alignment and ordering alignment. Extensive experiments on three ordinal classification tasks, including facial age estimation, historical color image (HCI) classification, and aesthetic assessment demonstrate its promising performance. The code is available at https://github.com/raywang335/L2RCLIP.

## 1 Introduction

Ordinal classification aims to predict labels that are related in a natural or implied order, which can be considered as a special case of ordinal regression after label discretization, i.e. discretize the continuous labels and each bin is then treated as a class. Common examples of such tasks are facial age estimation (e.g., estimating the facial age from 1 to 100), historical color image classification (e.g., assigning a time period to color photographs, ranging from the 1930s to the 1970s), and aesthetics assessment(e.g., rating image quality on a scale from "unacceptable" to "exceptional").

Compared with common classification, ordinal property of labels need to be additionally considered in ordinal classification. Many algorithms [4, 35, 12] employ the ordinal classification framework, which trains a set of classifiers or integrates probabilistic priors to directly predict rank labels. However, these methods exhibit suboptimal performance as a result of insufficiently harnessing the ordering properties. Order learning algorithms [48, 39, 26, 12] demonstrate competitive performance by effectively capturing relative ordering relationships. These approaches determine the target order of a

---

[\*]Corresponding author

37th Conference on Neural Information Processing Systems (NeurIPS 2023).

novel instance by contrasting it with well-defined reference instances. Nonetheless, the algorithm's performance can be substantially compromised by inadequate indexing quality of the reference instances.

Furthermore, many metric learning techniques [3, 41, 13, 49] have been developed to construct an ordinal embedding space in which the distances between diverse features effectively represent the differences in their respective ranks. However, all these methods learn ranking concepts depending solely on training data, which renders them vulnerable to overfitting[29].

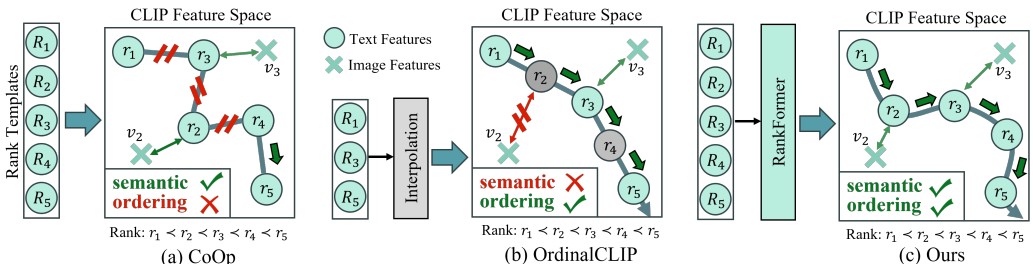

Figure 1: Comparison with CoOp(a) and OrdinalCLIP(b), where $v$ represents the image features and $R, r$ represents the rank templates in word embedding space and CLIP feature space, respectively. (a) CoOp aligns each rank template with its corresponding images via contrastive loss but vanilla CLIP fails to ensure ordering alignment. (b) OrdinalCLIP considers additional interpolation to explicitly maintain ordering alignment. However, the interpolation term can not preserve semantic alignment in CLIP feature space. (c) Our method enhance the ordering relation of vanilla rank templates while ensuring the semantic alignment of CLIP space.

Fortunately, recent developments in large pre-trained vision-language models offer new insights for various visual tasks [32, 27, 29, 25, 56]. Compared with visual content, human language contains highly abstract concepts and rich semantic knowledge [29, 32]. Motivated by it, we attempt to borrow knowledge from language domain by two major observations. Firstly, rank templates inherently contains ordinal information, e.g. *"a sixty years old face"* ≻ *"a ten years old face"*, which is also demonstrated in Figure 3(a). Secondly, inspired by [25, 56], rank features encapsulate the average information of numerous image features within a well-aligned cross-modal feature space, which can be considered as a robust prior for cross-modal metric learning.

Hence, we propose L2RCLIP to boost learning-to-rank of CLIP-based models for ordinal classification. Specifically, we first introduce a complementary prompt tuning method, termed RankFormer, to enhance the ordering relation of original rank templates. Specifically, RankFormer employs a token-wise attention layer and performs residual-style prompt blending with the original templates for prompt tuning. Moreover, inspired by pairwise metric learning [2], we propose cross-modal ordinal pairwise loss to ensure both semantic and ordering alignment in the CLIP feature space. Concretely, we revisit the approximate bound optimization of conventional cross-entropy loss and reformulate it in the cross-modal embedding space with ordinal language priors. OrdinalCLIP [29] also incorporates language priors to model ordering alignment, demonstrating impressive performance. However, our method distinguishes itself from previous works, as depicted in Figure 1. CoOp achieves semantic alignment through contrastive loss but fails to maintain ordering alignment. OrdinalCLIP addresses ordering alignment at the cost of weakened semantic alignment. Conversely, by leveraging Rank-Former and cross-modal ordinal pairwise loss, our approach simultaneously considers both semantic alignment and ordering alignment.

The contributions of this paper can be summarized as follows: (1) We incorporate learning-to-rank into vision-language pre-training model for ordinal classification, in which we present RankFormer to enhance the ordering relation of vanilla language prompts. (2) We explicitly utilize the language priors and further propose cross-modal ordinal pairwise loss to refine CLIP embedding space, in which image features and text features maintain both semantic and ordering alignment. (3) Extensive experiments demonstrate the competitive performance of L2RCLIP on age estimation, aesthetics assessment and historical image dating, as well as improvements in few-shot and distribution shift experiments.

## 2   Related Work

**Ordinal Classification**   Ordinal classification attempts to solve classification problems in which *not all wrong classes are equally wrong*. Early techniques [4, 45] adopted the classification framework and train a set of classifiers to directly estimate the rank labels. These methods got degraded performance due to ignoring the ordering relation. By incorporating probabilistic priors, Geng *et al.* [12] firstly proposed label distribution learning and assigned a Gaussian or Triangle distribution for an instance. The mean-variance loss was introduced in [39] for learnable label distribution and penalizes the learned variance of estimated distribution to ensure a sharp distribution. Probabilistic embedding [28] was developed to model the data uncertainty for ordinal regression. Liu *et al.* [33] proposed predicting the ordinal targets that fall within a certain interval with high confidence. These methods learn better rank concepts and significantly reduce the model's overconfidence toward incorrect predictions. In contrast to them, our L2RCLIP only focuses on enhancing vision-language alignment without complicated probabilistic distribution assumption.

Furthermore, many techniques [3, 41, 13, 49, 23, 24, 22] solve the ordinal classification task from the perspective of metric learning. These methods exploit the pairwise ordering relation in embedding space, where the distance between different features reflects ordinal information. For example, Ordinal log-loss (OLL) was presented in [3] with an additional distance-aware weighting term for ordinal classification. RankSim [13] proposed a regularization loss to ensure the ordinal consistency between label and feature. Suárez *et al.* [49] ranked the embedding distances between pairwise rank differences of features. Another way for ordinal classification is order learning [31, 48, 19], which learns ordering relation by comparison between instances. It usually show more promising results as learning relative ordering relation is much easier than learning absolute ordering relation. Lim *et al.* [31] firstly proposed this concept and determined the ordinal information of an unseen instance by compared to some known reference instances. Lee&Kim *et al.* [19] improved the quality of indexing to boost the performance. MWR [48] further proposed to learn a continuous regression score from reference instance. However, these methods often depend solely on training data to learn rank concepts, potentially leading to overfitting. To mitigate these issues, we leverage rich priors in human language to learn language-driven rank concepts.

**Vision-language Learning**   Vision-language pre-training(VLP) has significantly improved the performance on many downstream tasks by text and image matching, including segmentation [43, 8], object detection [14], image retrieval [36, 1], generation tasks [18, 40, 25, 5] and ordinal regression [29]. CLIP [42] and ALIGN [15] proposed to embed images and texts into the same representation space with two separate encoders in a contrastive-based approach. The experimental results show that the impressive "zero-shot" performance on downstream tasks, which show the power of language prior. Inspired by the recent advances in NLP, prompts and adapter-based tuning becomes prevalent in improving ability of CLIP. CLIP-Adapter [11] adds a light-weight module on top of image and text encoder. CoOp [55] proposes to learn context prompts for image classification. Due to lack of ordinal property, these methods lead to degraded performance in ordinal classification. Considering the great potential of language prior, we propose to incorporate learning-to-rank into CLIP for ordinal classification.

## 3   Proposed Approach

### 3.1   Problem Formulation

Ordinal classification is a unique case of image classification where labels possess ordinal properties. Mathematically, let $x_i \in \mathcal{X}$ denote the $i$-th input instance with $i = 1, 2, ..., N$, $y_i \in \{r_1, r_2, ..., r_M\}$ with ordered ranks $r_M \succ r_{M-1} \succ \cdots \succ r_1$ denote the ground truth value and $\hat{y}_i$ represent the predicted rank by the network, where $N$ represents the total number of instances, $M$ represents the number of ranks, and $\succ$ indicates the ordering between different ranks. Analogous to normal classification, ordinal classification aims to recover $y_i$ by encoding the image to feature $z_i = \Phi(x_i)$ with encoder $\Phi$ and then using a classifier $f_\omega(\cdot)$ to compute probability $p_i$. The predicted label $\hat{y}_i$ is the result with the highest probability $p_{y_i}$. The classification probability can be calculated by:

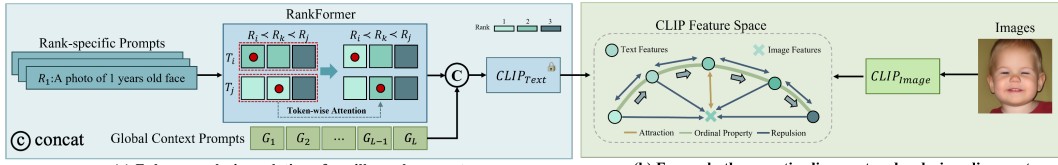

(a) Enhance ordering relation of vanilla rank prompts    (b) Ensure both semantic alignment and ordering alignment

Figure 2: An overview of the proposed L2RCLIP. We incorporate learning-to-rank into CLIP from two perspectives. First, RankFormer performs a token-level attention mechanism to enhance the ordering relation of vanilla rank prompts. Then, refined rank-specific prompts and randomly initialized context prompts are concatenated in the word embedding space and are sent to a text encoder to extract the corresponding text features. Moreover, we present two types of losses to refine CLIP feature space by attraction and repulsion, respectively. Attraction refers to an asymmetrical contrastive loss and a tightness term to attract paired images and text features while repulsion refers to a reweighting diversity term to ensure the ordering alignment.

$$p_i = \frac{exp(\omega_i^\top z_i)}{\sum_{j=1}^{M} exp(\omega_j^\top z_i)}. \tag{1}$$

To exploit ordinal information in language, ordinal classification can be transformed into a vision-language alignment task. Specifically, we use a pre-trained CLIP image feature extractor $Image(\cdot)$ to extract features from input images: $v_i = z_i = Image(x_i)$. For text features, we first construct hard rank templates $R = \{R_1, R_2, ..., R_M\}$ for a given ordinal classification task. For each template, we convert it into fixed-length tokens and then map them into 512-dimensional word embeddings. The language feature extractor $Text(\cdot)$ encodes the embeddings as a classifier weight $r_i$. The process can be formulated as: $r_i = w_i = Text(Tokenizer(R_i))$. Finally, we can calculate the prediction probability $p_i$ for rank $i$ with Eq.(1).

## 3.2 Proposed Method

Our objective is to integrate learning-to-rank into CLIP and learn language-driven rank concepts for ordering alignment while preserving original semantic alignment. Inspired by [16, 27], we firstly learn rank concepts in the word embedding space and subsequently refine the CLIP feature space to maintain both semantic and ordering alignment. Specifically, we introduce RankFormer to enhance the ordering relation of the original language prompts. RankFormer employs a token-wise attention layer for rank prompt tuning. To better utilize language prior, we reformulate the approximate bound optimization within cross-modal embedding space. Furthermore, we propose a cross-modal ordinal pairwise loss to ensure ordering alignment. Moreover, randomly initialized global context prompts and an asymmetrical contrastive loss are adopted to ensure semantic alignment. The overall framework is illustrated in Figure 2.

**Rank-specific Prompts.** As shown in Figure 3(a), rank templates in the vanilla CLIP contain some degree of ordinal information. However, a substantial proportion, nearly half of the pairs, lack clear ordinality. One intuitive approach to enhance this ordinal information is to further fine-tune these rank templates. However, implementing such a strategy introduces two primary challenges. First, the performance of different ranks for ordinal classification varies significantly driven by the imbalanced training data, and certain ranks cannot even be trained due to insufficient training data in extreme cases. Second, the contrastive loss introduces difficulties in enhancing the ordering relation during the training [53]. To address the first challenge, OrdinalCLIP [29] trains only a small number of base rank templates and generates other rank templates through explicit interpolation, which confines the ordinal information of vanilla rank templates. In this paper, we train a RankFormer with token-wise attention to enhance the ordinal information of the fixed rank templates. Specifically, we denote tokenized rank templates as $R \in \mathbf{R}^{M \times n \times D}$, where $M$ and $D$ represent the number of templates and word embedding channels, respectively. Note that we only consider the ranking tokens with length of $n$. Subsequently, we perform token-wise attention for prompt tuning and apply residual-style prompt blending with the original prompts. The process can be formulated as: $R^{'} = (1 - \alpha) \cdot R + \alpha \cdot f_{FFN}(f_{MSA}(f_{LN}(R)))$, where $\alpha$ represents the residual ratio.

**Cross-modal Ordinal Pairwise Loss.** Since cross-entropy loss ignored the ordinal information during training [53], we turn to a solution that recovers the ordering relation while maintaining semantic alignment. Inspired by the pairwise metric learning, we firstly revisit the vanilla cross-entropy loss from the perspective of approximate bound optimization. Boudiaf *et al.* [2] proved that minimizing the cross-entropy loss accomplishes by approximating a lower bound pairwise cross-entropy loss $L_{PCE}$. $L_{PCE}$ contains a tightness term and a diversity term, as follows:

$$L_{PCE} = \underbrace{-\frac{1}{2\lambda N^2} \sum_{i=1}^{N} \sum_{j:y_j=y_i} z_i^\top z_j}_{TIGHTNESS} + \underbrace{\frac{1}{N} \sum_{i}^{N} log \sum_{k=1}^{K} exp\left(\frac{\sum_{j=1}^{N} p_{jk} z_i^\top z_j}{\lambda N}\right) - \frac{1}{2\lambda} \sum_{k=1}^{K} ||c_k||}_{DIVERSITY}, \quad (2)$$

where $z_i$ represents the image feature in the embedding space, $p_{ij}$ represents the softmax probability of point $z_i$ belonging to class $j$, $c_k = \sum_{i=1}^{N} p_{ik} z_i$ represents the soft mean of class $k$ and $\lambda \in \mathcal{R}$ is to make sure $L_{CE}$ is a convex function with respect to encoder $\Phi_w$.

By incorporating the language priors, we turn Eq.(2) to a cross-modal pairwise cross-entropy loss. Specifically, as human language contains rich prior knowledge, text features can be considered as both hard mean $r_k = \frac{1}{N_k} \sum_{i=1}^{N_k} v_k$ and soft mean $r_k = \frac{1}{N} \sum_{i=1}^{N} p_{ik} v_i$ of image features at class $k$, where $N_k$ represents the sample number of class $k$ and $N$ represents the total sample number. Then, we reformulate the original pairwise cross-entropy loss $L_{PCE}$ as cross-modal pairwise cross-entropy loss $L_{CPCE}$:

$$L_{CPCE} = \underbrace{-\frac{1}{2\lambda N} \sum_{i=1}^{N} v_i^\top r_{y_i}}_{TIGHTNESS} + \underbrace{\frac{1}{N} \sum_{i}^{N} log \sum_{k=1}^{K} exp\left(\frac{v_i^\top r_k}{\lambda}\right) - \frac{1}{2\lambda} \sum_{k=1}^{K} ||r_k||}_{DIVERSITY}, \quad (3)$$

Inspired by [53], we use meanNN entropy estimator [10] to estimate the diversity term in Eq.(3). The process is formulated as:

$$L_{CPCE}^{diversity} \propto \frac{D}{N(N-1)} \sum_{i=1}^{N} \sum_{j\neq i}^{N} log(v_i^\top r_j + r_i^\top r_j), \quad (4)$$

$$L_{CPCE}^{tightness} \propto -\frac{1}{N} \sum_{i=1}^{N} v_i^\top r_{y_i}, \quad (5)$$

where $D$ represents the feature dimensions. To recover ordering relation, we propose an additional weighting term. Intuitively, each rank template will have a high similarity score with images of a close rank and a low similarity score with images of a distant rank. As such, we opt to weight the cross-modal features with $w_{ij}$, where $w_{ij}$ are the distances in the label space. The final cross-modal ordinal pairwise loss is defined as follows:

$$L_{cop}(y_i) = \frac{1}{(B-1)} \sum_{j\neq i}^{B} w_{y_i j} \cdot (v_{y_i} + \gamma r_{y_i})^\top r_j - v_{y_i}^\top r_{y_i}, \quad (6)$$

where $\gamma$ controls the strength of the rank within rank templates and $B$ represents the batchsize. To further refining the CLIP feature space, we also propose a simplified cross-modal ordinal pairwise loss $L_{scop}$ with language-related parameters frozen(i.e. $\gamma = 0$).

**Global Context Prompts.** Given that global context prompts significantly surpass manually designed discrete prompts in vision tasks [55, 27, 54], we integrate them with our complementary rank-specific prompts in RankFormer to enhance semantic alignment. Specifically, we randomly initialize $L$ global context prompts, denoted as $G = \{G_1, G_2, ..., G_L\}$, and concatenate them with rank-specific prompts in the word embedding space of CLIP.

**Asymmetrical Contrastive Loss.** In vanilla CLIP [42], models are optimized using the standard contrastive loss, including a text-image contrastive loss $L_{t2i}$ and an image-text contrastive loss $L_{i2t}$. In this work, we replace the original contrastive loss with an asymmetrical contrastive loss due to

many-to-many image-text mappings within a batch. In other words, different images in a batch may have the same rank, and the same rank may have the same description. Therefore, we replace $t_i$ with $t_{y_i}$ in both text-image contrastive loss $L_{t2i}$ and image-text contrastive loss $L_{i2t}$. Specifically, our improved asymmetrical contrastive loss is defined as follows:

$$L_{t2i}(y_i) = \frac{1}{|Z(y_i)|} \sum_{z \in Z(y_i)} log \frac{exp(v_z^\top r_{y_i}/\tau)}{\sum_{j=1}^{B} exp(v_j^\top r_{y_i}/\tau)}, \tag{7}$$

where $\tau$ is the temperature parameter and $Z(y_i) = \{z \in 1...B : y_z = y_i\}$.

### 3.3 Loss functions

We employ a two-stage training scheme to refine our approach following [27]. In the first stage, our focus is to enhance the ordinal information of the vanilla rank templates. We utilize cross-modal ordinal pairwise loss $L_{cop}$ and the asymmetrical contrastive losses $L_{t2i}$ and $L_{i2t}$. In the second stage, our objective is to simultaneously improve both semantic alignment and ordering alignment within the CLIP latent space. We use the simplified cross-modal ordinal pairwise loss $L_{scop}$ and the cross-entropy loss $L_{ce}$. More detail can be found in Sec. 4.1 .

## 4 Experiments

### 4.1 Implementation details

We adopt the ViT-B/16 visual encoder and the text encoder from CLIP [42] as the backbone for our image and text feature extractor. All training data is resized to 224 × 224 and subjected to random horizontal flipping. For all experiments, we employ the Adam [17] optimizer with default settings. The learning rates for the RankFormer and the visual backbone are $3.5 \times 10^{-4}$ and $1 \times 10^{-5}$, respectively. The model is trained for 20 epochs in the first stage and 40 epochs in the second stage, with the learning rate decayed by a factor of 0.1 in epoch 30. We set $L = 5$ in most of our experiments. For MORPH and CLAP2015, we set $L_{t2i}$, $L_{i2t}$, and $L_{cop}$ with weights of 0.03, 0.03, and 3 for the first stage, and $L_{ce}$ and $L_{scop}$ with weights of 1 and 1 for the second stage. For other datasets, we set $L_{t2i}$, $L_{i2t}$, and $L_{cop}$ with weights of 0.1, 0.1, and 3 for the first stage, and $L_{ce}$ and $L_{scop}$ with weights of 1 and 1 for the second stage. All experiments are conducted on a single NVIDIA 3090 GPU.

### 4.2 Age Estimation

Table 1: Results on MORPH II and CLAP2015.

| Methods | Morph MAE(↓) | CLAP2015 MAE(↓) |
|---|---|---|
| AGEn [50] | 2.52 | 2.94 |
| BridgeNet [30] | 2.38 | 2.87 |
| AVDL [52] | 2.37 | - |
| POE [28] | 2.35 | - |
| DRC-ORID [19] | 2.26 | - |
| PML [6] | 2.15 | 2.91 |
| MWR [48] | 2.13 | 2.77 |
| Vanilla CLIP [42] | 6.91 | 4.66 |
| CoOp [55] | 2.39 | 2.75 |
| OridinalCLIP [29] | 2.32 | — |
| L2RCLIP(Ours) | **2.13** | **2.62** |

Table 2: Results on Adience dataset.

| Methods | Adience Accuracy(%, ↑) | Adience MAE(↓) |
|---|---|---|
| OR-CNN [37] | 56.7 | 0.54 |
| CNNPOR [34] | 57.4 ± 5.8 | 0.55 ± 0.08 |
| GP-DNNOR [35] | 57.4 ± 5.5 | 0.54 ± 0.07 |
| SORD [7] | 59.6 ± 3.6 | 0.49 ± 0.05 |
| POE [28] | 60.5 ± 4.4 | 0.47 ± 0.06 |
| MWR [48] | 62.6 | 0.45 |
| GOL [20] | 62.5 | 0.43 |
| Vanilla CLIP [42] | 43.3 ± 3.6 | 0.80 ± 0.02 |
| CoOp [55] | 60.6 ± 5.5 | 0.50 ± 0.08 |
| OridinalCLIP [29] | 61.2 ± 4.2 | 0.47 ± 0.06 |
| L2RCLIP(Ours) | **66.2 ± 4.4** | **0.36 ± 0.05** |

**Datasets.** Age estimation aims to predict the age of a given facial image. We train and evaluate our method on the widely-used MORPH II [44] dataset, CLAP2015 [9] dataset, and Adience [21] dataset. MORPH II [44] is one of the largest and most commonly used longitudinal face databases and we follow the widely adopted training and evaluation following [45, 28, 48, 47]. CLAP2015 [9] is used for apparent age estimation, where each image is rated by at least 10 annotators, and the mean rating

is set as the ground truth. Following [48], we split it into 2,476 for training, 1,136 for validation, and 1,079 for testing. Adience [21] contains discrete labels annotated with eight age groups. We adopt the same five-fold cross-validation protocol used in [21]. For evaluation metric, we use the mean average error (MAE) to measure the absolute differences between the ground truth labels and the predicted ones. Classification accuracy is additionally adopted for Adience. For detailed experimental settings, please refer to the supplementary material.

**Comparison with State-of-the-art Methods.**    In Table 1, our L2RCLIP outperforms both conventional algorithms and language-powered algorithms in all tests. For results on the Morph II, compared with conventional algorithms, L2RCLIP achieves state-of-the-art performance on MAE at 2.13, verifying the significance of leveraging rich priors in human language. Furthermore, compared to methods utilizing language priors, our L2RCLIP exhibits substantial performance improvements. CoOp significantly enhances the vanilla CLIP's performance by learning global context prompts for semantic alignment. Subsequently, OrdinalCLIP further reduces MAE by 0.07 through fixed interpolation for ordering alignment. Nonetheless, a considerable margin remains compared to our methods, which validates the effectiveness of our approach.

Table 1 also compares the results on CLAP2015. Due to the great challenge, many previous methods adopt additional boosting schemes [30, 50]. However, without using such schemes, L2RCLIP outperforms all conventional algorithms. Compared with language-guided models, our L2RCLIP also shows much better results on MAE, specifically a significant MAE margin of 0.15 evaluated on the test set. Table 2 shows the comparison results on Adience [21] using the metrics of MAE and Accuracy. Compared with Morph II and CLAP2015, Adience is used for age group estimation. The underline indicates the mean value reported in the original paper. Our method outperforms the state-of-the-art algorithms by significant gaps of 5.7% in accuracy and 0.07 in MAE.

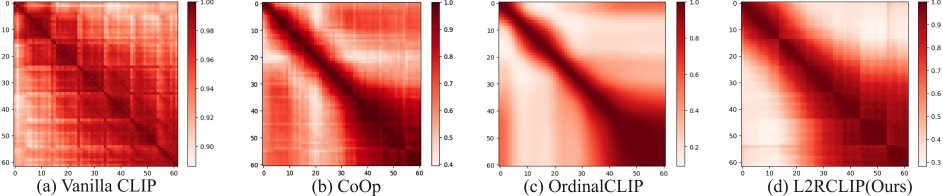

(a) Vanilla CLIP          (b) CoOp          (c) OrdinalCLIP          (d) L2RCLIP(Ours)

Figure 3: The similarity matrices of rank templates in different methods. The redder, the more similar the pair of rank templates. The percentages of templates pairs that obey the ordinality are: 55.36%, 59.92%, 65.94%, and **71.87%**, respectively.

**Ordinality of Learned Rank Templates.**    Following [29], we report the ordinality score by measuring the cosine similarity between rank templates. The ordinality score is the percentage of rank template pairs that obey the ordinal property. Figure 3 reports the ordinality scores in different methods. Vanilla CLIP in Figure 3(a) contains a certain degree of ordering relations, where more than half of the rank template pairs show correct ordering relation. Moreover, CoOp in Figure 3(b) introduces additional global context prompts and improves the ordinality score by 4.56%. OrdinalCLIP in Figure 3(c) adopts an explicit interpolation strategy and further improves the ordinality by 6.02%. However, there is a noticeable "red stripe" in the upper right corner, indicating that this section of rank template pairs significantly violates ordinality. We believe that explicit interpolation effectively ensures local ordinal properties, but may fail to preserve global ordinal properties. Our methods greatly alleviate this problem and achieve a higher ordinality score. The "red blob" in the lower right corner results from insufficient samples for old individuals. Therefore, to avoid the model's overconfidence towards incorrect predictions, this part of rank templates should exhibit higher similarity while ensuring ordering alignment. By comparison, our method outperforms previous methods.

**Few-shot Learning.**    Table 4 demonstrates the generalization ability of L2RCLIP for few-shot learning. Following [29], the full dataset is split into 80% for training and 20% for testing. The entire test set is used for validation, and only 1/2/4/8/16/32/64 samples in the training set from each class of labels are chosen for training. We observe that introducing rank information in CLIP

---

i.e., $OS(\%) = \sum_{i=1}^{M} \sum_{j=i}^{M-1} \mathbb{I}\{s_{i,j} > s_{i,j+1}\}/(M \times (M-1)/2)$, where $M$ is the number of templates and $s_{i,j}$ represents the cosine similarity of each pair of templates.

Table 3: We report the MAE results under few-shot settings on the MOPRH II dataset.

| # Shots | 1 | 2 | 4 | 8 | 16 | 32 | 64 |
|---|---|---|---|---|---|---|---|
| CoOp [55] | 5.09 | 4.50 | 3.81 | 3.57 | 3.23 | 2.87 | 2.61 |
| OrdinalCLIP [29] | 4.94 | 4.36 | 3.55 | 3.31 | 3.07 | 2.76 | 2.57 |
| L2RCLIP(Ours) | **4.54** | **3.92** | **3.40** | **3.28** | **2.81** | **2.55** | **2.38** |

actually benefits few-shot learning tasks. Compared with CoOp, both OrdinalCLIP and our L2RCLIP achieve significant improvements across all settings, which verifies the effectiveness of ordering alignment. Moreover, our L2RCLIP further reduces the average MAE by 0.24, which demonstrates the effectiveness of the proposed learning-to-rank method in CLIP.

Table 4: The MAE results under the distribution shift setting on the MOPRH II. "re cls" denotes the number of reduced classes, and "re smp" means the percentage of reduced sampled in one class.

| re cls - re smp | 10-80 | 10-90 | 20-80 | 20-90 | 30-80 | 30-90 | 40-80 | 40-90 |
|---|---|---|---|---|---|---|---|---|
| CoOp [55] | 2.71 | 2.85 | 2.98 | 3.51 | 3.06 | 3.36 | 2.99 | 3.30 |
| OrdinalCLIP [29] | 2.61 | 2.67 | 2.77 | 3.06 | 2.86 | 3.21 | 2.84 | 3.12 |
| L2RCLIP(Ours) | **2.28** | **2.30** | **2.37** | **2.43** | **2.51** | **2.61** | **2.68** | **2.79** |

**Distribution Shift.** Following [29], we conduct data distribution shift experiments on the MORPH II dataset for generalization. We use the same setting as the general regression setting. For the training set, we randomly choose several rank labels, e.g., 10, 20, 30, and 40. Then, in those classes, we randomly discard some portion of training data, e.g., 80 and 90. We report our experiments in Table 4. For the most severe settings, CoOp, OrdinalCLIP, and our L2RCLIP methods incur performance losses of 42.24%, 34.49%, and 30.98%, respectively, which demonstrates that our approach exhibits superior robustness when faced with data distribution shift problems. For all shift settings, our method shows better performance than OrdinalCLIP, which highlights the effectiveness of the proposed learning-to-rank method in CLIP compared with interpolation.

Table 5: Ablation Study of L2RCLIP on the MORPH II and CLAP2015.

| | Ablation | Choices | | | | | | | |
|---|---|---|---|---|---|---|---|---|---|
| | RankFormer | | ✔ | | | | ✔ | ✔ | ✔ |
| | $L_{cop}$ | | | ✔ | | ✔ | ✔ | | ✔ |
| | $L_{scop}$ | | | | ✔ | ✔ | | ✔ | ✔ |
| Morph II | MAE(↓) | 2.50 | 2.48 | 2.37 | 2.38 | 2.21 | 2.29 | 2.31 | **2.13** |
| | OS(%,↑) | 55.89 | 57.58 | 66.15 | 55.89 | 66.15 | 71.87 | 57.58 | **71.87** |
| CLAP2015 | MAE(↓) | 2.75 | 2.68 | 2.71 | 2.71 | 2.70 | 2.65 | 2.66 | **2.62** |
| | OS(%,↑) | 53.33 | 54.77 | 57.33 | 53.33 | 57.33 | 67.55 | 55.25 | **67.55** |

## 4.3 Analysis

**Ablation Study.** In this section, we conduct an ablation study to examine the respective roles of each component in L2RCLIP, and the results are reported in Table 5. Three key observations can be drawn from Table 5. First, each proposed component demonstrates improvements over the baseline model and complements one another. Second, although RankFormer marginally enhances the ordinal information of vanilla rank templates, its performance remains unsatisfactory compared to other methods when only contrastive loss is utilized for training. This result further highlights the importance of our cross-modal pairwise loss in refining the CLIP feature space. Third, fine-tuning rank templates without RankFormer significantly impairs performance on CLAP2015, potentially due to imbalanced training for certain rank templates. This finding underscores the effectiveness of RankFormer in modeling the ordering relation derived from the complete set of rank templates.

**Initialization Impact.** As we aim to enhance the ordering relation vanilla rank templates, it is essential to examine the effects of initialization. We report our results in Table 6. Although our

Table 6: Initialization Impact Analysis.

| Initialization | MAE($\downarrow$) | OS($\%$, $\uparrow$) |
|---|---|---|
| *A photo of {age} years old face.* | 2.13 | 71.87 |
| *Age estimation: a person at the age of {age}.* | 2.14 | 72.07 |
| *The age of the person in the portrait is {age}.* | 2.12 | 71.32 |
| *The age of the person is {age}.* | 2.16 | 71.52 |
| *The age of the face is {age}.* | 2.14 | 71.08 |
| Mean/Std | 2.14/0.01 | 71.57/0.40 |

methods fix the original rank templates, different initializations lead to similar convergence and performance with a low standard deviation value of 0.014. This observation further substantiates the robustness of our methods against diverse initializations.

Table 7: Results on Morph, CLAP2015 and Adience datasets.

| Methods | Morph (MAE, $\downarrow$) | CLAP2015 (MAE, $\downarrow$) | Adience (Accuracy, $\uparrow$) | Adience (MAE, $\downarrow$) |
|---|---|---|---|---|
| L2RCLIP-I | 2.19 | 2.78 | $62.9 \pm 5.5$ | $0.42 \pm 0.06$ |
| L2RCLIP (Ours) | **2.13** | **2.62** | **$68.2 \pm 7.2$** | **$0.36 \pm 0.05$** |

**Compared with interpolation-based method.** To further prove the effectiveness of our proposed method, we compare our L2RCLIP with previous interpolation-based method (e.g. OrdinalCLIP[29]). We adopt the same setting except the process of ordinality learning and we term it as L2RCLIP-I. We report the results on the aging dataset. More experiments for few-shot learning and distribution shift and the implementation details setting for L2RCLIP-I can be found in appendix.

As illustrated in Table 7, our method outperforms interpolation-based methods with a significant margin in experiments involving a large number of rank categories. This outcome is attributable to the challenge posed by direct interpolation methods in modelling complex ordering relationships. Our approach continues to surpass interpolation-based methods even in experiments with a smaller number of rank categories. Collectively, these experiments corroborate the effectiveness of the methods proposed in this study.

### 4.4 Image Aesthetics Assessment

**Datasets.** CrowdBeauty [46] consists of 13,929 available Flickr photos across four categories: nature, animal, urban, and people. The aesthetic quality of each image is evaluated using five absolute rating scales: "unacceptable", "flawed", "ordinary", "professional", and "exceptional". Following previous methods [29, 20], we select 80% of the images for training and the rest for testing. Five-fold cross-validation is employed for fair comparisons. Both the mean MAE and accuracy are reported.

Table 8: Results on Image Aesthetics dataset.

| Methods | Accuracy($\%$, $\uparrow$) | | | | | MAE($\downarrow$) | | | | |
|---|---|---|---|---|---|---|---|---|---|---|
| | Nature | Animal | Urban | People | Overall | Nature | Animal | Urban | People | Overall |
| CNNPOR [34] | 71.86 | 69.32 | 69.09 | 69.94 | 70.05 | 0.294 | 0.322 | 0.325 | 0.321 | 0.316 |
| SORD [7] | 73.59 | 70.29 | 73.25 | 70.59 | 72.03 | 0.271 | 0.308 | 0.276 | 0.309 | 0.290 |
| POE [28] | 73.62 | 71.14 | 72.78 | 72.22 | 72.44 | 0.273 | 0.299 | 0.281 | 0.293 | 0.287 |
| GOL [20] | 73.8 | 72.4 | 74.2 | 69.6 | 72.7 | 0.27 | 0.28 | 0.26 | 0.31 | 0.28 |
| Vanilla CLIP [42] | 65.24 | 45.67 | 58.78 | 53.06 | 55.68 | 0.461 | 0.557 | 0.468 | 0.524 | 0.502 |
| CoOp [55] | 72.74 | 71.46 | 72.14 | 69.34 | 71.42 | 0.285 | 0.298 | 0.294 | 0.330 | 0.302 |
| OrdinalCLIP [29] | 73.65 | 72.85 | 73.20 | 72.50 | 73.05 | 0.273 | 0.279 | 0.277 | 0.291 | 0.280 |
| L2RCLIP(Ours) | 73.51 | **75.26** | **77.76** | **78.69** | **76.07** | **0.267** | **0.253** | **0.216** | **0.246** | **0.245** |

**Results.** Table 8 presents our results on the CrowdBeauty dataset. Aesthetic score prediction is challenging due to the subjectivity and ambiguity of aesthetic criteria; however, our L2RCLIP demonstrates state-of-the-art performance on most experimental settings by fully exploring learning-to-rank with language prior. Compared with OrdinalCLIP, L2RCLIP improves accuracy by 3.02% and

reduces MAE by 0.35 overall, which further verifies the effectiveness of our proposed learning-to-rank method. When compared with the best approach without language prior, L2RCLIP improves overall MAE by 0.35 and overall accuracy by 3.37%. Consistent improvements are observed across all categories compared to previous methods, showcasing the effectiveness of our proposal in exploiting language ordinal information.

### 4.5 Historical Image Dating

**Datasets.** The historical image dating dataset [38] serves as a benchmark for automatically predicting the decade of historical colored images. The dataset comprises five-decade categories, ranging from the 1930s to the 1970s. Following [34, 38], we adopt the same train-test split and ten-fold cross-validation. Both the mean and standard deviation for MAE and accuracy metrics are reported.

**Results.** Table 9 showcases improvements compared to other state-of-the-art models using the full dataset. Initially, zero-shot CLIP exhibits suboptimal performance resulting from inadequate semantic and ordinal alignment. CoOp enhances the results by incorporating global context prompts to facilitate semantic alignment. Furthermore, Ordinal-CLIP exploits interpolation to improve the average MAE by 0.09. Our L2RCLIP further advances the average MAE by 0.25 with a lower standard deviation. In comparison with the previous conventional model, L2RCLIP achieves a new state-of-the-art performance with an MAE of 0.43 and an accuracy of 67.22%, thereby validating the effectiveness of our proposed method.

Table 9: Results on HCI dataset.

| Methods | HCI | |
|---|---|---|
| | MAE($\downarrow$) | Accuracy($\%, \uparrow$) |
| CNNPOR [34] | $0.82 \pm 0.05$ | $50.12 \pm 2.65$ |
| GP-DNNOR [35] | $0.76 \pm 0.05$ | $46.60 \pm 2.98$ |
| POE [28] | $0.76 \pm 0.04$ | $54.68 \pm 3.21$ |
| MWR [48] | $\underline{0.58}$ | $\underline{57.8}$ |
| GOL [20] | $\underline{0.55}$ | $\underline{56.2}$ |
| Vanilla CLIP [42] | $1.01 \pm 0.03$ | $30.41 \pm 3.32$ |
| CoOp [55] | $0.76 \pm 0.06$ | $51.94 \pm 2.60$ |
| OridinalCLIP [29] | $0.67 \pm 0.03$ | $56.44 \pm 1.66$ |
| L2RCLIP(Ours) | $\mathbf{0.43 \pm 0.02}$ | $\mathbf{67.22 \pm 1.59}$ |

## 5 Discussions and Conclusions

In this paper, we propose L2RCLIP to boost learning-to-rank of CLIP for ordinal classification. Specifically, we introduce a complementary prompt tuning method, termed RankFormer, to enhance the ordering relation of the original rank prompts. It performs token-level attention and residual-style prompt blending for prompt tuning. Additionally, we revisit the approximate bound optimization of cross-entropy and reformulate it in the cross-modal embedding space by incorporating the language knowledge. To additionally recover the ordinal information, we further introduce cross-modal ordinal pairwise loss to refine the ordering alignment of CLIP feature space. Extensive experiments demonstrate the effectiveness of our approach on various ordinal classification tasks, including facial age estimation, historical image dating, and image aesthetics assessment. Furthermore, L2RCLIP outperforms in the challenging few-shot learning and data distribution shift learning scenarios. Lastly, we conduct a comprehensive ablation study to verify the effectiveness of each component of our proposed method.

**Broader Impacts.** As a versatile approach, L2RCLIP can be applied to any ordinal classification tasks such as image aesthetics assessment or other rank assessment. However, these tasks may pose a risk of unlawful surveillance or invasion of privacy if abused. Meanwhile, as L2RCLIP is based on a large-scale vision-language model, addressing demographic biases in pre-trained vision-language models is of significant importance. Therefore, we emphasize that L2RCLIP represents a research proof of language-driven learning and is not appropriate for real-world usage without strict technical controls.

**Acknowledgement.** This research is sponsored by National Natural Science Foundation of China (Grant No. 62306041, 62006228), Beijing Nova Program (Grant No. Z211100002121106, 20230484488, 20230484276), and Youth Innovation Promotion Association CAS (Grant No.2022132).

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

# 6 Appendix

This supplementary material begins with a comprehensive visualization of the datasets central to our study. The specifics of our experimental settings are subsequently outlined in Section 6.2. Section 6.1 features an expanded analysis, including results from ablation studies. A key highlight of this section is the visual interpretation of the CLIP image features facilitated by t-SNE [51]. Concurrently, a comparative analysis is conducted, comparing the efficacy of interpolation-based strategies with our learning-based methods(i.e. L2RCLIP).

## 6.1 More Analysis of L2RCLIP

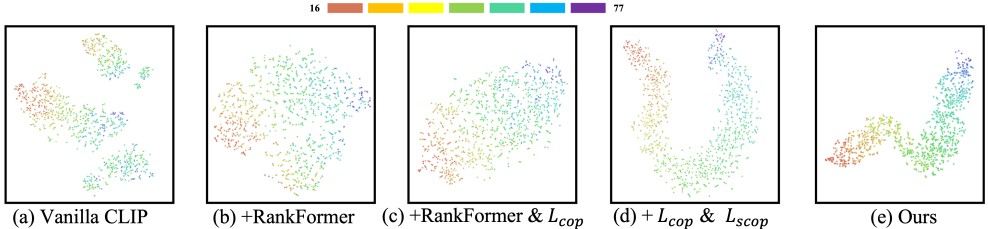

| (a) Vanilla CLIP | (b) +RankFormer | (c) +RankFormer & $L_{cop}$ | (d) + $L_{cop}$ & $L_{scop}$ | (e) Ours |

Figure 4: Visualizing the ablation effects on the MORPH II dataset: t-SNE visualizations of 512D spaces in CLIP latent space.

**Additional Ablation Study.** Figure 4 presents the embedding spaces corresponding to various ablation settings, mapped from the original 512D embedding spaces via t-SNE [51]. In Panel (a), the vanilla CLIP implementation reveals a sub-optimal ordering relationship among images of distinct ranks. Feature demarcations across different ranks are ambiguous, displaying considerable overlap. The incorporation of RankFormer with global context prompts [55], illustrated in Panel (b), aids in improving and consolidating the order alignment of these image features. As shown in Panel (c), the implementation of our proposed loss function, $L_{cop}$, further enhances this order alignment. Conversely, Panel (d) indicates that without the RankFormer module, the order alignment between the lowest and highest ranks falls short of the desired outcome. These findings substantiate the efficacy of the modules proposed in this study, with each component playing a significant role in the overall model performance. This underscores the criticality of their synergistic implementation.

Table 10: Batchsize Analysis.

| Batchsize | 16 | 32 | 64 | 96 | 128 |
|---|---|---|---|---|---|
| MAE($\downarrow$) | 2.15 | 2.13 | 2.13 | 2.17 | 2.17 |
| OS(%, $\uparrow$) | 67.21 | 68.55 | 71.87 | 71.71 | 75.42 |

We also explore the role of different batchsize settings. We report the result in Table 10. According to the result, we choose batchsize with 64 to conduct other experiments.

Table 11: The MAE results under the distribution shift setting and few shot setting on the MOPRH II.

| re cls - re smp | 10-90 | 20-80 | 20-90 | 30-80 | 30-90 | 40-80 | 40-90 |
|---|---|---|---|---|---|---|---|
| L2RCLIP-I | 2.39 | 2.45 | 2.50 | 2.57 | 2.70 | 2.73 | 2.93 |
| L2RCLIP(Ours) | **2.30** | **2.37** | **2.43** | **2.51** | **2.61** | **2.68** | **2.79** |
| #Shots | #1 | #2 | #4 | #8 | #16 | #32 | #64 |
| L2RCLIP-I | **4.31** | 4.02 | 3.63 | 3.48 | 3.13 | 2.80 | 2.62 |
| L2RCLIP(ours) | 4.54 | **3.92** | **3.40** | **3.28** | **2.81** | **2.55** | **2.38** |

**Compared with Interpolation-based method.** In this study, we also propose an interpolation technique for our basic rank templates, which we term L2RCLIP-I. Two distinguishing characteristics set L2RCLIP-I apart from OrdinalCLIP [29]. Firstly, we utilize the ViT-B/16 visual backbone of

CLIP for image feature extraction, whereas OrdinalCLIP employs a pre-trained VGG-16 network supplemented by a linear projection layer. Secondly, our method relies on a two-stage training strategy, in contrast to the one-stage approach adopted by OrdinalCLIP. Importantly, both training strategies require a comparable time commitment. The results are reported in Tables 7 and 11.

**Local ordinality score of L2RCLIP.** To further prove our L2RCLIP have learned better ordering relationship, we follow OrdinalCLIP[29] and use the local ordinality score. The formula is defined as: $LOS(\%) = \sum_{i=1}^{K} \sum_{j=i}^{K} \mathbb{I}\{s_{i,j} > s_{i,j+1}\}/(K \times (K-1)/2)$, where $K$ is the size of local window and $s_{i,j}$ represents the cosine similarity of each pair of templates. We propose that the locally linear manifold can be preserved within a fixed small window size. Therefore, we calculate the local ordinality score using window sizes of 2, 4, 8, 16, and 32. The results of the local ordinality score are shown in Table 12.

Table 12: The local ordinality score results on the MORPH II dataset.

| # window size | 2 | 4 | 8 | 16 | 32 |
|---|---|---|---|---|---|
| Vanilla CLIP | 100.00 | 83.33 | 78.57 | 70.83 | 60.08 |
| OrdinalCLIP[29] | 100.00 | 100.00 | 100.00 | 96.19 | - |
| L2RCLIP(Ours) | **100.00** | **100.00** | **100.00** | **100.00** | **97.78** |

Table 13: Ablation study of global context prompts and architecture.

| Method | Morph(MAE) | Morph(OS%) | CLAP2015(MAE) | CLAP2015(OS%) |
|---|---|---|---|---|
| Vanilla CLIP | 6.91 | 55.36 | 4.66 | 52.51 |
| w/o context prompt | 2.23 | 65.46 | 2.76 | 67.17 |
| RankFormer→ MLP | 2.27 | 67.48 | - | - |
| L2RCLIP(Ours) | **2.13** | **71.87** | **2.62** | **67.55** |

**More ablation study of L2RCLIP** To avoid the effect of token mixing and the type of architecture of RankFormer, we conduct more detailed ablation study. The results are shown in Table13. Our proposed method is complementary to previous prompt tuning methods and our L2RCLIP can achieve comparable performance with OrdinalCLIP even without context prompt. We have also compared L2RCLIP performance with an MLP-based architecture to avoid effects driven by extra computation. Note that both RankFormer and MLP have similar training parameters. The results show that our token-wise RankFormer can enhance the ordinality between input rank templates.

Table 14: Additional results on Morph II.

| Methods | Setting A | Setting B | Setting C | Setting D |
|---|---|---|---|---|
| DRC-ORID[19] | 2.26 | 2.51 | 2.58 | 2.16 |
| OL[31] | 2.41 | 2.75 | 2.68 | 2.22 |
| MWR-G[48] | 2.24 | 2.55 | 2.61 | 2.16 |
| GOL[20] | 2.17 | 2.60 | **2.51** | 2.09 |
| L2RCLIP(Ours) | **2.13** | **2.53** | 2.56 | **1.95** |

**More results on Morph II datasets** We have conducted experiments on the other three settings of Morph II. The results are presented in Table 14. The details for each settings are as follows:

- Setting A: A total of 5,492 images of Caucasians are sampled and then randomly divided into training and test sets with a ratio of $8 : 2$.

- Setting B: Approximately 21,000 images of Caucasians and Africans are randomly selected, ensuring a balanced ratio of $1 : 1$ between Caucasians and Africans, as well as a ratio of $1 : 3$ between females and males. The dataset is then divided into three subsets (S1, S2, S3). The training and testing process is repeated twice: 1) training on S1 and testing on S2+S3, and 2) training on S2 and testing on S1+S3.

- Setting C: The entire dataset is randomly partitioned into five folds, with the constraint that images of the same person belong to only one fold. The 5-fold cross-validation is then performed.
- Setting D: The entire dataset is randomly divided into five folds without any restrictions. The 5-fold cross-validation is then performed.

## 6.2 Experiment settings

**Dataset Details.** In the scope of this study, we only utilize publicly available data. To provide a comprehensive understanding of the tasks at hand, we illustrate a selection of random samples from the image aesthetics assessment dataset (Figure 5) and the historical image dating dataset (Figure 6). To further enhance our exposition, Figure 7 depicts both the original and adjusted distributions of the MORPH II dataset.

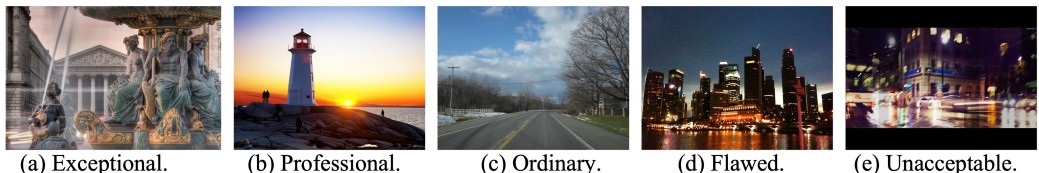

(a) Exceptional.     (b) Professional.     (c) Ordinary.     (d) Flawed.     (e) Unacceptable.

Figure 5: Samples from the urban collections of the aesthetics dataset.

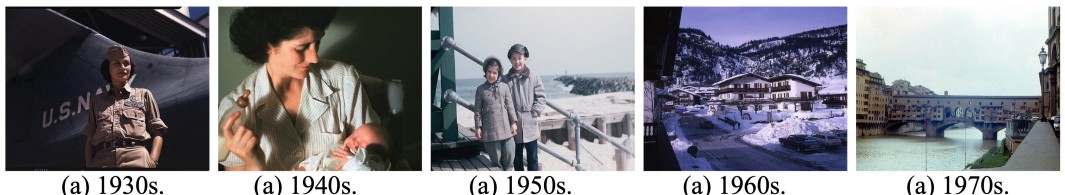

(a) 1930s.     (a) 1940s.     (a) 1950s.     (a) 1960s.     (a) 1970s.

Figure 6: Samples from the historical image dating dataset.

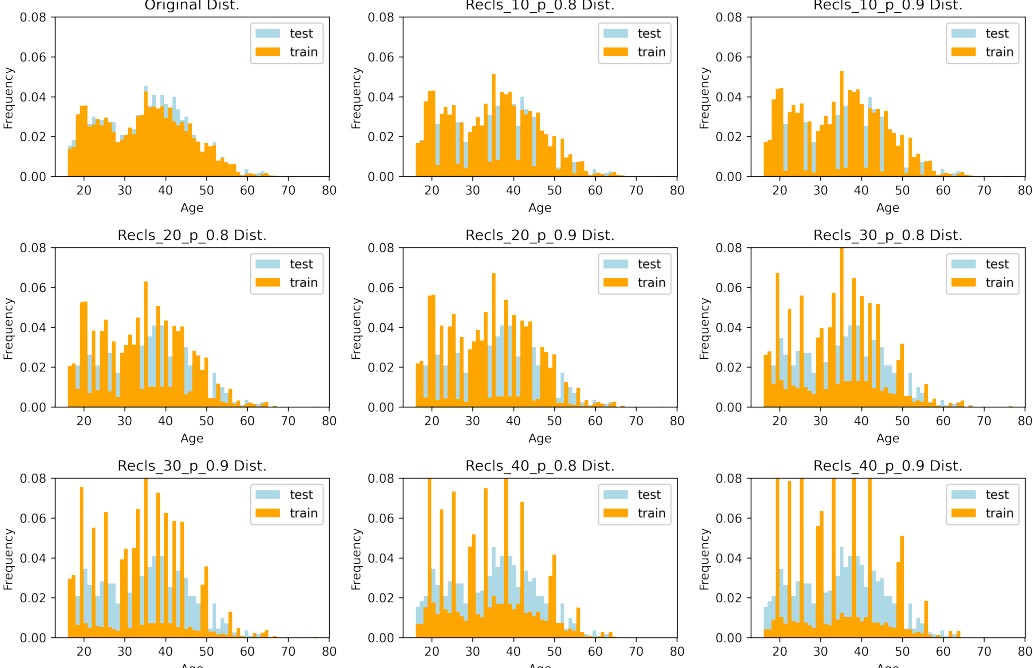

Figure 7: Original and shifted distributions of the MORPH II dataset.

