# OpenReview forum: "Learning-to-Rank Meets Language: Boosting Language-Driven Ordering Alignment for Ordinal Classification"
_NeurIPS.cc/2023/Conference — NeurIPS 2023 poster_

### Official Review · Reviewer_v94e · 2023-07-05

**Soundness:** 3 good
**Presentation:** 3 good
**Contribution:** 3 good
**Rating:** 6
**Confidence:** 2

**Summary:**

The paper presents a novel language-driven ordering alignment method called L2RCLIP for ordinal classification. The authors leverage pre-trained vision-language models to incorporate rich ordinal priors from human language. They propose RankFormer, a prompt tuning technique that enhances the ordering relation of rank prompts using token-level attention and residual-style prompt blending. Additionally, they introduce a cross-modal ordinal pairwise loss to refine the CLIP feature space, ensuring semantic and ordering alignment between texts and images. The proposed method is evaluated on facial age estimation, historical color image classification, and aesthetic assessment tasks, showing promising performance.


**Strengths:**

1. The paper introduces a novel method that leverages language priors to address the overfitting issue in ordinal classification.
2. The experimental results indicate that the proposed method achieves promising performance on the evaluated tasks, suggesting its effectiveness in addressing the overfitting problem.


**Weaknesses:**

N/A

**Questions:**

N/A

---

> ### Author Rebuttal · Authors · 2023-08-09
>
> # Response to the Reviewer v94e
> Comment:
> Thank you for your positive review and constructive feedback.

---

### Official Review · Reviewer_6B9T · 2023-07-05

**Soundness:** 2 fair
**Presentation:** 2 fair
**Contribution:** 2 fair
**Rating:** 7
**Confidence:** 5

**Summary:**

The paper presents L2RCLIP, a novel language-driven ordering alignment method for ordinal classification. The authors propose to leverage the rich ordinal priors in human language by converting the original task into a vision-language alignment task. The method introduces a complementary prompt tuning technique called RankFormer, designed to enhance the ordering relation of original rank prompts. In addition, the authors propose a cross-modal ordinal pairwise loss to refine the CLIP feature space, where texts and images maintain both semantic alignment and ordering alignment. The method is evaluated on three ordinal classification tasks, including facial age estimation, historical color image (HCI) classification, and aesthetic assessment, showing promising performance.


**Strengths:**

The proposed loss function is an interesting contribution to the field, as it provides a new perspective on viewing cross-entropy loss within the context of ordinal regression. The authors' approach to incorporating language priors and restructuring the cross-modal embedding space using cross-modal ordinal pairwise loss is innovative and well-presented.
The method demonstrates impressive performance on various ordinal classification tasks, outperforming previous state-of-the-art methods. This indicates the potential of L2RCLIP in addressing real-world problems related to ordinal classification, including facial age estimation, historical image dating, and image aesthetics assessment.


**Weaknesses:**

The comparison with previous methods seems unfair, as the authors use a much stronger image backbone from CLIP, while previous papers use VGG16 as their backbone. This could be a significant factor contributing to the improved performance of L2RCLIP. It would be beneficial for the authors to provide a fair comparison by also evaluating their method using a similar backbone to previous works.
In Figure 3, some unexpected spikes around age 10 and some cube-like patterns are observed, indicating that the ordinality score is not as smooth as expected, despite being higher than previous methods. The authors could consider reducing the window of comparisons and evaluating the ordinality score more comprehensively. This would provide a better understanding of the method's performance in terms of ordinality.
The claim of Rank-specific prompts being effective in enhancing the ordering relation is too strong, and there is no experimental evidence to support it. While token mix may increase computation and information flow, it does not necessarily guarantee the ordinal property of rank features or the final textual features. The authors should provide experimental evidence or a more detailed explanation to support this claim.
The paper could be improved by describing the final loss composition in the main text, which is currently missing. Providing a clear explanation of how the different components of the loss function are combined would help the reader better understand the proposed method.


**Questions:**

Can the authors provide a fair comparison by evaluating L2RCLIP using a similar backbone to previous works?
How can the ordinality score be better evaluated to provide a more comprehensive understanding of the method's performance?
Could the authors provide experimental evidence or a more detailed explanation to support the claim of Rank-specific prompts being effective in enhancing the ordering relation?
Can the authors describe the final loss composition in the main text, providing a clear explanation of how the different components of the loss function are combined?

**Limitations:**

The authors need to analysis the limitations of their work.

---

> ### Author Rebuttal · Authors · 2023-08-09
>
> # Response to Reviewer 6B9T
> We sincerely appreciate your positive review and valuable comments. Please find our responses below.
> ***
> **Q1: Compred with previous method with the same architecture**
>
> **[Reply]** Our method doesn't show promising result using VGG16, which may be reasonable since our two key designs based on the well-aligned text-image latent space. For fairness, we keep the same experiment setting and retrain the interpolation-based method based on OrdinalCLIP. The results and training details are as follows:
>
> **Table 1. Results on Morph, CLAP2015 and Adience datasets**
> |Method| Morph (MAE) | CLAP2015 (MAE) | Adience (MAE) | Adience (Acc.)
> |---|---|---|---|---|
> |L2RCLIP-I|2.19|2.78| 0.42 ± 0.06|62.9 ± 5.5|
> |L2RCLIP(ours)|**2.13**|**2.62**|  **0.36 ± 0.05** |**66.2±4.4**|
>
> **Table 2. The MAE results under the distribution shift setting on the MOPRH II**
> |re cls - re smp| 10-90| 20-80| 20-90| 30-80| 30-90| 40-80| 40-90|
> |---|---|---|---|---|---|---|---|
> |L2RCLIP-I| 2.39| 2.45 |2.50 |2.57| 2.70| 2.73| 2.93|
> |L2RCLIP(ours)| **2.30**| **2.37**|**2.43**|**2.51** |**2.61**| **2.68**| **2.79**|
>
> **Table 3. The MAE results under few shot setting on the MOPRH II**
> |#shots |#1| #2| #4| #8 |#16| #32| #64|
> |---|---|---|---|---|---|---|---|
> |L2RCLIP-I| **4.31** |4.02| 3.63 |3.48| 3.13| 2.80| 2.62|
> |L2RCLIP(ours)| 4.54| **3.92** |**3.40** |**3.28**| **2.81**| **2.55**| **2.38**|
>
> For fairness, we use the official code of OrdinalCLIP for interpolation. We conduct three groups of experiment to verify the effectiveness of  our methods. **First**, as illustrated in Table 1, our method outperforms interpolation-based methods with a significant margin in experiments involving a large number of rank categories. This outcome is attributable to the challenge posed by direct interpolation methods in modelling complex ordering relationships. **Second**, our methods  exhibits superior performance in the majority of few-shot learning tasks and distribution shift tasks, when compared to interpolation-based methods. Collectively, these experiments corroborate the effectiveness of the methods proposed in this study.
>
> **The detail for L2RCLIP-I**: Firstly, we utilize the ViT-B/16 visual backbone of CLIP for image feature extraction, whereas OrdinalCLIP employs a pre-trained VGG-16 network supplemented by a linear projection layer. Secondly, our method relies on a two-stage training strategy, in contrast to the one-stage approach adopted by OrdinalCLIP.
>
> **Q2: The ordinality score with less ranks.**
>
> **[Reply]** Thanks for your suggestions. We reduce the number of ranks for better visualization. **The results are shown in the .pdf file.**
>
> **Q3: The ordinal property learned by RankFormer.**
>
> **[Reply]** To avoid token mixing effect, we conduct global context prompt ablation study. The results are as follows.
>
> **Table 4. Ablation study of global context prompts.**
> |Method	|Morph (MAE)|	Morph (OS)|	CACD (MAE)|	CACD (OS)|
> |---|---|---|---|---|
> |Vanilla CLIP	|6.91	|55.36%|	4.66|	52.51%|
> |CoOp(Variant)	|2.39	|59.92%|	2.75	|53.33%|
> |w/o context prompt|	2.23|	65.46%	|2.76	|67.17%|
> |L2RCLIP(Ours)|	2.13|	71.87%	|2.62	|67.55%|
>
> The above result have shown that our porposed methods actually help to learn ordinal property. Additionally, we visualize the embedding space by t-SNE. **The results are shown in the .pdf file.**
>
> **Q4: The training detail about final loss used in L2RCLIP**
>
> **[Reply]** Due to page limitations, we have included this section in the supplementary materials. We utilize the cross-modal ordinal pairwise loss $L_{cop}$ and asymmetrical contrastive loss $L_{t2i}$ and $L_{i2t}$ to learn reliable rank prompts. In the second stage, we employ the cross-entropy loss $L_{ce}$ and simplified cross-modal ordinal pairwise loss $L_{scop}$ to fine-tune the image encoder. **Further details can be found in Supp. Line 17-25.**
>
> Every effort has been made to address your comments faithfully in the revised paper. If you have any additional comments, please let us know. Thank you again for your positive and insightful comments. We do appreciate them.

---

> > ### Comment · Reviewer_6B9T · 2023-08-14
> >
> > Thank you very much for your detailed explanation of my concerns.
> >
> > I should restate my view that the paper presents a novel approach to ordinal classification and shows promising performance on various tasks. However, the comparison with previous methods seems unfair, and some claims lack supporting evidence. Addressing these concerns and providing a more comprehensive evaluation of the ordinality score would strengthen the paper and justify a higher rating.
> >
> > Till now, I am satisfied with the authors' reply, apart from:
> > 1) The use of rank token transformer does not guarentee the ordinality. In fact, I personally do not deem the linear comparison to be the true ordinality, since the high-dimensional statistics are highly complex. Therefore, the t-SNE may not be strong enough to support the claim but is suitable for validating the intuition. Anyway, the performance is indeed improved with more computation, so I kindly suggest the authors to rephrase their depiction about the rank token interaction improving ordinality, otherwise I would thought it is over-claimed.
> > 2) It could be better to compare the ordinality score at a small range of value space, instead of merging the ranks.
> >
> > Thanks again for your informative reply and I would like to here more updates from the authors.

---

> > > ### Author Response · Authors · 2023-08-15
> > > **Response to Reviewer 6B9T**
> > >
> > > Thanks for your reply. We would like to clarify the issues as follows.
> > > ***
> > > **Q1. Ordinality score and the ordinlaity learned by RankFormer**
> > >
> > > **[Reply]** **For ordinality score**, We acknowledge that this metric is not perfect because it is challenging to maintain the linear assumption in a high-dimensional manifold as you suggested. To alleviate this strong assumption, we propose that the locally linear manifold can be preserved within a fixed small window size. Therefore, we calculate the local ordinality score using window sizes of 2, 4, 8, 16, and 32. The results of the local ordinality score are shown in Table 1. The results demonstrate that our L2RCLIP effectively maintains local ordinality within a fixed small window size. **For ordinality learned by RankFormer**. We believe that transformer-based architectures are more capable of modeling the relationships between input tokens. In ordinal classification tasks, the training model may tend to leverage ordinal information as it is a straightforward method to minimize loss. Based on this assumption, we propose a token-wise RankFormer to enhance the ordinality between input rank templates.
> > > We have also compared its performance with an MLP-based architecture to avoid effects driven by extra computation. The results are presented in Table 2. Note that both RankFormer and MLP have similar training parameters. **Finally**, we sincerely appreciate your valuable advice. We will make sure to revise our content regarding the ordinality in the upcoming version to enhance rigor and clarity.
> > >
> > >
> > >
> > >
> > > Table 1. The local ordinality score results on the MORPH II dataset.
> > > |#window size|	#2|	#4|	#8	|#16	|#32|
> > > |---|---|---|---|---|---|
> > > |Vanilla CLIP|	100.00%	|83.33%|	78.57%	|70.83%	|60.08%|
> > > |OrdinalCLIP	|100.00%	|100.00%	|100.00%	|96.19%	|—|
> > > |L2RCLIP(Ours)	|100.00%	|100.00%	|100.00%	|100.00%	|97.78%|
> > >
> > > Table 2. Ablation study on architecture of proposed models.
> > > |Arch.	|OS	|MAE|
> > > |---|---|---|
> > > |MLP	|67.48%|	2.27|
> > > |RankFormer	|71.87%	|2.13|
> > >
> > > **Q2. The merging of ranks**
> > >
> > > **[Reply]** We would like to clarify any potential misunderstanding. We do not merge any ranks; instead, we only choose 6/12/20 ranks from the left corner in Fig. 3 for better visualization. Our primary focus is to describe relative ordinality rather than absolute ordinality, so we apply maxmin normalization within each local window.
> > >
> > > Please let us know if it addresses your concern. Thank you again for your insightful comments. We do appreciate them.

---

> > > > ### Comment · Reviewer_6B9T · 2023-08-16
> > > >
> > > > Thanks for your reply and all my concerns have been addressed. I am inclined to raise my rating to a clear accept.

---

### Official Review · Reviewer_XrDm · 2023-07-09

**Soundness:** 3 good
**Presentation:** 2 fair
**Contribution:** 2 fair
**Rating:** 6
**Confidence:** 3

**Summary:**

In this paper, a novel ordinal regression framework based on CLIP is proposed. The proposed algorithm, which is called L2RCLIP, exploits language priors together with image features.
It encourages that image features at each class locate around the text feature of that class in the embedding space. To this end, L2RCLIP uses RankFormer to obtain text features and CLIP image encoders to obtain image features.
The network is optimized via cross-modal ordinal pairwise loss. Extensive experiments on various ordinal regression tasks show that the proposed algorithm outperforms the previous SOTA, ordinalCLIP.


**Strengths:**

1. The paper is easy to follow. It describes the proposed algorithm clearly and seems like to be reproducible.

2. The proposed algorithm is simple but technically sound. It also achieves the best scores in most tests.

3. Experiments are diverse and solid enough to evaluate the performances properly. It also provides extensive ablation studies for better understanding on each part of the proposed algorithm



**Weaknesses:**


* Major
1. It would be better to discuss about the many loss functions in Eq(2)~Eq(6) more deeply. It lacks the how the loss function operates to encourage network training to be performed into the desirable direction.

2. Are global context prompts learnable parameters as well?

3. If so, it would be interesting to see the ablation result for global context prompts. It may be used as the auxiliary role for the language priors, but it may reduce the impact of the rank template encoding.
In such a case, without the global context prompts, the results in Table 6 may be changed meaningfully.

4. MORPH II has 4 widely used evaluation settings. In the paper, the evaluation on the most simple setting is provided only. It would be helpful to compare the performances on the other challenging settings.

* Minor

1. L135, Fig. 3.2 -> Figure 2

2. In Eq (1):  z_j -> z_i?

**Questions:**

Please see the weakness section for my questions on the paper. In overall, I'm leaning to accept because the proposed algorithm is clearly described, and technically sound. Also, it achieves the good scores on various benchmark tests. However, I will see the other reviewer's opinion and the author response too.

**Limitations:**

The authors have addressed the potential negative social impact in the main paper but I was not able to find specific discussion about the limitation.

---

> ### Author Rebuttal · Authors · 2023-08-09
>
> # Response to Reviewer XrDm
> We sincerely appreciate your positive review and valuable comments. Please find our responses below.
> ***
>
> **Q1: To prove the effectiveness of proposed losses**
>
> **[Reply]** Thank you for your suggestion. We provide a detailed analysis of Eq(2) to Eq(6) as follows:
>
> - **Eq(2)->Eq(3)** Compared to images, language contains a higher density of information. Intuitively, the information conveyed by **"a 20-year-old person"** and a large number of images containing 20-year-old people is similar. Therefore, we attempt to combine language features with pairwise loss in a mean form.
>
> - **Eq(3)->Eq(4)** Previous work has addressed the diversity term through meanNN-based entropy estimation.
>
> - **Eq(3)->Eq(5)** The tightness term can be straightforwardly transformed into a loss objective function.
>
> - **Eq(4),Eq(5)->Eq(6)** We introduce a simple distance term to further enhance the ordering relation.
>
>
> **Q2: The role of global context prompts**
>
> **[Reply]** The results of the related ablation study are presented in Table 1:
>
> **Table 1.  Ablation study of global context prompts.**
> |Method	|Morph (MAE)	|Morph (OS)	|CACD (MAE)	|CACD (OS)|
> |  ----  | ----  | ----  | ----  | ----  |
> |Vanilla CLIP	|6.91	|55.36%	|4.66	|52.51%|
> |CoOp(Variant)	|2.39	|59.92%	|2.75	|53.33%|
> |w/o context prompt	|2.23	|65.46%	|2.76	|67.17%|
> |L2RCLIP(Ours)	|2.13	|71.87%	|2.62	|67.55%|
>
> **Q3: Results on all settings of Morph II**
>
> **[Reply]** We have conducted experiments on the other three settings of Morph II. The results are presented in Table 2.
>
> **Table 2. Additional results on Morph II**
> |Method	|SettingA	|SettingB	|SettingC	|SettingD|
> |  ----  | ----  | ----  | ----  | ----  |
> |MWR-G(2022, CVPR)	|2.24	|2.55	|2.61	|2.16|
> |GOL(2022, NeurlPS)	|2.17	|2.60	|**2.51**	|2.09|
> |L2RCLIP(Ours)	|**2.13**	|**2.53**	|2.56	|**1.95**|
>
> **Q4: Some minor typos error**
>
> **[Reply]** Thank you for pointing out the typos. We have corrected the mentioned errors in our revised paper.
>
> Every attempt has been made to address your comments faithfully in the revised paper. If you have any additional comments, please let us know. Thank you again for your positive and insightful comments. We do appreciate them.

---

### Official Review · Reviewer_yU8S · 2023-07-13

**Soundness:** 2 fair
**Presentation:** 1 poor
**Contribution:** 2 fair
**Rating:** 3
**Confidence:** 4

**Summary:**

The paper proposed to leverage vision-and-language models to improve ordinal classification. This is a follow-up work on the previous OrdinalCLIP paper. The major contribution is RankFormer, which is designed to enhance the ordering of the original rank prompts. Also a cross-modal ordinal pairwise loss is proposed to refine the CLIP feature space. Experimental results are presented on three ordinal classification tasks, including facial age estimation, historical color image classification, and aesthetic  assessment.

**Strengths:**

The intuition behind the proposed method makes sense to me. The overall idea is simple and straightforward, and should be easy to reproduce. The experimental results seem to be extensive and can demonstrate the effectiveness of the proposed method.


**Weaknesses:**

The following are more detailed comments and suggestions about the paper.

1, In Line 10-11, if the goal is to incorporate language priors, why use the CLIP model? The text encoder in CLIP is not very strong. Existing LLMs can provide much better language priors.

2, The paper claims that the proposed model is designed to learn both semantic and ordering based on Figure 1. It is better to provide some examples or analysis about how the semantic and ordering are learned simultaneously. The visualization in Figure 3 is not very helpful.

3, The paper may want to provide more details about RankFormer in Line 145-152. Why call it token-wise attention? It seems to be the basic attention mechanism. Maybe illustrate the architecture of RankFormer as well.

4, In Figure 2, why fix the CLIP_{Text} encoder but fine tuning the CLIP_{Image} encoder? The “C” in the left part means “concatenation”?

5, In Line 147-149, “k is the length of rank templates”, so k is smaller than M? How to pick k in this formulation?

6, The writing of the paper can be much better:
In Line 149, is the the length of rank templates…
Where is Fig. 3.2 in Line 135?
“k” is used everywhere in the paper: in Line 182, for the number of global context prompts; in Figure 2 for the index of rank templates; in Line 149, for the length of rank templates.

7, In Line 184, the paper proposed to use asymmetrical contrastive loss to handle many-to-many image-text mapping within the batch. It is unclear to me why this asymmetrical loss can better handle many-to-many mapping? Please elaborate more.

8, As a follow-up paper of OrdinalCLIP, the paper is an incremental improvement over OrdinalCLIP, and the novelty of the paper seems to be limited.

**Questions:**

See above.

**Limitations:**

See above.

---

> ### Author Rebuttal · Authors · 2023-08-09
>
> # Response to Reviewer GKDm
> We would like to thank the reviewer for the valuable comments. However, we feel there is some misunderstanding. We clarify the issues and address the questions accordingly as described below.
> ***
> **Q1: Choice of using powerful language model**
>
> **[Reply]** That may be a promising research direction to further improve performance in ordinal classification. Due to limited computational resources, we have chosen the CLIP text encoder to provide language priors. Despite this limitation, we have found that CLIP performs well in three tasks of ordinal classification: age estimation, historical color image classification, and aesthetic assessment. Our proposed method has achieved state-of-the-art (SOTA) performance on several test benchmarks. As you suggested, we believe our method can provide new insights for further LLM-based ordinal classification.
>
> **Q2: Validation of learned ordinal property and semantic alignment**
>
> **[Reply]** **For ordinal property**, we use three kinds of method to verify it learned by our proposed modules. **First**, we follow OrdinalCLIP(Li et al, NeurlPS2022) and adopt ordinality score to measure the distance of normalized rank templates quantitatively and qualitatively. We outperform the previous method by over 5.93%. **Second**, we conduct comprehensive ablation study on two different datasets to verify the effectiveness of our proposed method quantitatively. **Finally**, we visualize the embedding space for the ablated methods in the Supp. Fig.4. We think these qualitative and quantitative analysis will support the rank information learned by our proposed method.
>
> **For semantic alignment**, we think the semantic alignment can be measured using metrics such as MAE when CLIP is adopted for classification. Our method performs the best in 15 out of 16 benchmark tests, which can be used to prove that our method use better semantic alignment. Moreover,   we conduct additional ablation study on global context prompts to further prove our RankFormer and proposed  cross-modal ordinal pairwise loss can achieve both semantic alignment and ordering alignment compared with previous method. The results are shown in Table 1.
>
> Table 1. Ablation study of global context prompts.
>
> |Method	|Morph (MAE)|	Morph (OS)	|CACD (MAE)|	CACD (OS)|
> |---|---|---|---|---|
> |Vanilla CLIP	|6.91	|55.36%	|4.66	|52.51%|
> |CoOp(Variant)	|2.39	|59.92%	|2.75|	53.33%|
> |OrdinalCLIP(w context prompt)	|2.32	|65.94%	|—	|—|
> |w/o context prompt	|2.23	|65.46%	|2.76	|67.17%|
> |L2RCLIP(Ours)	|2.13	|71.87%	|2.62	|67.55%|
>
>
>
>
> **Q3: Explaination of token-wise attention in RankFormer**
>
> **[Reply]** Given an input tensor $x\in R^{M\times N\times C}, the normal attention operates on the second dim while token-wise attention on the first dim since we want to enhance the ordinal property in the vanilla rank prompts. In fact,  RankFormer handles three different types of tokens. **First**, for special tokens like [EOS], RankFormer keeps them the same during training. These special tokens are not optimized. **Second**, for normal tokens, RankFormer functions similarly to linear layers. **Lastly**, for rank tokens, RankFormer employs a token-level attention mechanism to further enhance the ordinal property. The detailed architecture of RankFormer will be included in our revised version.
>
> **Q4: Training parameters in CLIP text encoders and image encoders**
>
> **[Reply]** Since we only have coarse rank templates, the performance of text encoders may be significantly degraded after full-parameter fine-tuning. Therefore, we only fine-tune a minimal number of parameters in the text branch. Additionally, "C" represents concatenation, and we will include this correction in our revised framework.
>
> **Q5: Explaination of meaning of notations**
>
> **[Reply]** Apologies for any confusion caused. In fact, $k$ here is usually smaller than the max token length (which is 77 in the most of case) in CLIP. $M$ is the number of rank templates or ordinal categories in ordinal classification, e.g. $M$=101 if the range of age estimation is [0,100]. We choose $k$ based on our pre-defined rank templates. We will exclude special token in practice.
>
> **Q6: Revision of some confusing typos**
>
> **[Reply]** We will ensure to avoid repeated notation in our revised paper. Thanks for your advice.
>
> **Q7: Explaination of asymmetrical loss**
>
> **[Reply]** In contrast to the normal contrastive learning in CLIP, where each image has only one target label, our cases involve images with multiple target labels within a batch. Directly adopting the symmetrical loss used in CLIP would be suboptimal in this scenario. We need to take into account all the target labels in a batch. Therefore, we group the correct matches in the similarity map and compute the loss by taking the mean.
>
> **Q8: Limited novelty compared with OrdinalCLIP**
>
> **[Reply]** We do not agree. The analysis is as follows:
> Both OrdinalCLIP and our L2RCLIP aim to leverage language priors for the ordinal classification task. The main difference lies in the design of the ranking mechanism. OrdinalCLIP manually designs interpolation rules and applies interpolation to a few learnable rank prompts, achieving good results on certain test benchmarks.
> However, we argue that this explicit interpolation may involve a tradeoff between semantic alignment and ordering alignment, as the interpolated result may not guarantee correct semantic alignment. In response to this, we have designed a new token-wise RankFormer and a novel cross-modal ordinal pairwise loss. This allows us to learn a more complex ranking mechanism while preserving semantic alignment. Our method has achieved SOTA) performance. Overall, our L2RCLIP outperforms in 15 out of 16 benchmark tests.
>
> Every attempt has been made to address your comments faithfully in the revised paper. If you have any additional comments, please let us know. Thank you again for your valuable comments. We do appreciate them.

---

> ### Author Response · Authors · 2023-08-18
> **Looking forward to the response from Reviewer yU8S**
>
> Dear Reviewer yU8S,
>
> We have tried our best to address all the concerns and provided as much evidence as possible. May we know if our rebuttals answer all your questions? We truly appreciate it.
>
> Best regards,
>
> Author #3203

---

> > ### Comment · Reviewer_yU8S · 2023-08-21
> >
> > Thank the authors for answering my questions during rebuttal. Most of my questions have been addressed during rebuttal. I will increase my rating of the paper to borderline accept.

---

> > > ### Author Response · Authors · 2023-08-21
> > > **Response to Reviewer yU8S**
> > >
> > > Dear Reviewer yU8S,
> > >
> > > We greatly appreciate that you will increase the score of our article. This response is just to **remind you that you may have forgotten to change the score in the OpenReview system**. We truly appreciate it.
> > >
> > > Best regards,
> > >
> > > Author #3203

---

### Official Review · Reviewer_GKDm · 2023-07-24

**Soundness:** 3 good
**Presentation:** 3 good
**Contribution:** 3 good
**Rating:** 5
**Confidence:** 3

**Summary:**

This paper proposes L2RCLIP, which features two modules for ordinal classification with vision-language models, i.e., CLIP. The first is a token-wise attention module called RankFormer to tune the rank prompts. And the second is a pairwise ordinal loss to inject rank information into the supervision. Synergically, the two modules achieves competitive performance across age estimation, aesthetics assessment and historical image dating benchmarks, as well as improvements in few-shot and distribution shift experiments.

**Strengths:**

1.	The proposed two modules are simple and effective in improving ordinal classification ability of CLIP models.
2.	The token-wise attention in prompt tuning is interesting.
3.	The writing of this paper is of good quality.



**Weaknesses:**

1.	Considering rank information in the loss has been a common practice in ordinal classification methods as depicted in the related works (line80-line92). It is unclear how language priors are applied in the proposed pairwise loss (Eq.6), and thus distinguish this loss from existing methods.
2.	Although the token-wise attention is new and intuitively incorporates the information of different rank prompts, the ordinal properties with strictly ordered ranks are not assured in the attention process.
3.	Some definition of variables may cause confusion, e.g., The ‘T’s in Eq.3,4,5 represent the output embeddings of text encoder while the ‘T’s in Figure2(a) seems representing the prompt embeddings. And it’s confusing that whether ‘k’s in line147 and line149 are for the same thing.


**Questions:**

1.	The authors might give qualitative or quantitative analysis of the rank prompts to support that rank information is learned and preserved by RankFormer.
2.	The authors might give more elaboration on the synergy of the two proposed modules in Table 5, e.g., explain what necessitates the use of a pairwise loss to unleash the power of RankFormer and how / whether the pairwise loss can be regarded as an indispensable part of the RankFormer?
3.	Some details need the authors’ clarification:
1)	Why OrdinalCLIP does not preserve semantic alignment as it has also learned a set of context prompts other than the rank prompts?
2)	As the rand templates in an Mxkxc tensor in the token-wise attention (line147), does the attention operate on the 1st dimension (M) or on the 2nd dimension (k)?
3)	What does “language-related parameters frozen” mean in line 178? Does it mean the prompting part (including context and rank prompts) is frozen or only the text encoder is frozen.
4)	As Lscop is a simplified special case of Lcop, how would the two co-exist in Table 5.



**Limitations:**

Yes

---

> ### Author Rebuttal · Authors · 2023-08-09
>
> # Response to Reviewer GKDm
> We thank the reviewer for the valuable feedback and a positive assessment of our work. We are happy the reviewer finds the paper well-organised and our method interesting, valuable, and innovative with good performance. Below we detail our response to the review concerns.
> ***
> **Q1: The qualitative and quantitative analysis of rank information**
>
> **[Reply]** We use three kinds of method to verify the rank information learned by our proposed modules. **First**, we follow OrdinalCLIP(Li et al, NeurlPS2022) and adopt **ordinality score** to measure the distance of normalized rank templates quantitatively and qualitatively. We outperform the previous method by over 5.93%. **Second**, we conduct **comprehensive ablation study** on two different datasets to verify the effectiveness of our proposed method. **Finally**, we **visualize the embedding space** for the ablated methods in the Supp. Fig.4. We think these qualitative and quantitative analysis will support the rank information learned by our proposed method.
>
> **Q2: Explaination of proposed two modules**
>
> **[Reply]** We aim to improve the performance of ordinal classification by focusing on two key aspects.
> **First**, through our experiments, we have observed that vanilla text prompts already possess a certain degree of ordinal property. Building upon this observation, we have designed a **token-wise RankFormer module** to further enhance the ordering alignment within these prompts. This module specifically focuses on capturing and reinforcing the correct ordering relationships between different tokens. **Second**, taking inspiration from previous work on metric learning, we have **incorporated language knowledge** into a lower bound of cross-entropy loss. Additionally, we have introduced **an additional distance weighting term** to effectively model the embedding space with better ordering alignment. This helps to ensure that the learned representations exhibit the desired ordinal properties.
>
> From the perspective of experiment, we have conducted a comprehensive ablation study. This study allows us to individually assess the impact of each module and examine how their effects can be combined. Furthermore, we have visualized the embedding space of each ablated model, providing additional insights into the behavior and performance of our proposed approach. These visualizations can be found in Supp. Fig. 4.
>
>
>
> **Q3.1: The semantic alignment in OrdinalCLIP by context prompts**
>
> **[Reply]** We understand your points regarding the semantic alignment of OrdinalCLIP. We agree that context prompts can enhance semantic alignment, as evidenced by previous works like CoOp and other prompt tuning methods.
>
> However, we would like to emphasize two important aspects. **First**, rule-based interpolation does not guarantee that the interpolated results will always adhere to the correct semantic alignment. This can potentially lead to suboptimal performance on downstream tasks.
> **Second**, we consider semantic alignment can be measured using metrics such as MAE when CLIP is adopted for classification. Our methods not only demonstrate better performance on several benchmark tests but also, as shown in Table 1, exhibit promising results in both semantic alignment and ordering alignment, even without the use of global context prompts.
>
> Table 1. Ablation study of global context prompts.
>
> |Method	|Morph (MAE)	|Morph (OS)	|CACD (MAE)	|CACD (OS)|
> |---|---|---|---|---|
> |Vanilla CLIP	|6.91	|55.36%	|4.66	|52.51%|
> |CoOp(Variant)	|2.39	|59.92%	|2.75	|53.33%|
> |OrdinalCLIP(w context prompt)	|2.32	|65.94%	|—	|—|
> |w/o context prompt	|2.23	|65.46%	|2.76	|67.17%|
> |L2RCLIP(Ours)	|2.13	|71.87%	|2.62	|67.55%|
>
>
> **Q3.2: Token-wise attention in RankFormer**
>
> **[Reply]** The attention mechanism in RankFormer operates on the first dimension. RankFormer handles three different types of tokens.
> **First**, for special tokens like [EOS], RankFormer keeps them the same during training. These special tokens are not optimized.
> **Second**, for normal tokens, RankFormer functions similarly to linear layers.
> **Lastly**, for rank tokens, RankFormer employs a token-level attention mechanism to further enhance the ordinal property.
>
> **Q3.3&Q3.4 : Training detail about language-related parameters and $L_{cop}$/$L_{scop}$**
>
> **[Reply]** We fix the text encoder and image encoder and only train global context prompts and RankFormer in the first stage and only finetune the image encoder in the second stage.  As you suggested, $L_{scop}$ is the special case of $L_{cop}$. We firstly use $L_{cop}$, $L_{t2i}$ and L_{i2t} to learn reliable rank prompts. Then, we use $L_{scop}$ and cross-entropy loss $L_{ce}$ to finetune the image encoder for better performance. **See more details in Supp Line 17-25.** In the second stage, all language-related parameters are frozen. So entropy estimation of last term in Eq(4) can be ignored, which is equivalent to setting $\lambda =0$. Hope the misunderstanding can be cleared by our explanation.
>
>
>
> **W1: The application of lanuage prior and novelty compare with previous method**
>
> **[Reply]** As far as we are aware, there have been limited studies that specifically focus on incorporating language priors into metric learning techniques. Compared with language-powered method, e.g. OrdinalCLIP, we propose RankFormer and cross-modal ordinal pairwise loss to boost performance of ordinal classification and achieve the SOTA performance on several test benchmarks.
>
> **W2: The learned ordinal properties by RankFormer**
>
> **[Reply]** We explain W2 carefully in Q1. Please refer to Q1 for more details.
>
> **W3: Some typos mistakes**
>
> **[Reply]** We will ensure to avoid repeated notation and correct any mistakes in our revised paper.
>
> Every attempt has been made to address your comments faithfully in the revised paper. If you have any additional comments, please let us know. Thank you again for your positive and insightful comments. We do appreciate them.

---

> > ### Comment · Reviewer_GKDm · 2023-08-21
> > **Thanks for the response**
> >
> > Thanks for your response. This addresses my questions.

---

### Official Review · Reviewer_VQC9 · 2023-07-27

**Soundness:** 3 good
**Presentation:** 2 fair
**Contribution:** 2 fair
**Rating:** 4
**Confidence:** 4

**Summary:**

This paper proposes a language-driven ordering alignment method for ordinal classification. For the language prompt, this paper introduces the RankFormer, which uses Transformer to learn token-wise attention over a set of rank templates. For the loss function, this paper presents a cross-modal ordinal pairwise loss under the pairwise cross-entropy loss formulation. Experimental results show the effectiveness of the proposed method.

**Strengths:**

1.	Considering language prior to ordinal regression is a promising direction.

2.	New SOTA is achieved as shown in the experiments.


**Weaknesses:**

1.	The design of RankFormer doesn’t make any sense. RankFormer is essentially learning R’ = f_W(R), where W and R are learnable parameters. As R’ is only conditional on W and R, learning R and R’ are completely equivalent mathematically. Any R’ learned by the proposed method is in the solution space of R. They are not fundamentally different, and any difference in performance between the two could be due to the randomness of the network.

2.	It’s unclear what is the total loss used in this paper.

3.	In Line 178, the meaning of “To further refine the CLIP feature space, we also propose a simplified cross-modal ordinal pairwise loss L_scop with language-related parameters frozen.” is unclear. What’s the purpose of L_scop? There is no significant difference between L_cop and L_scop. Why can’t merge them and simply use one Loss? Mathematically, you can achieve the same effect by changing the value of \lambda.

4.	Line 187: “Many to many image-text mappings within a batch”. No, it’s one-to-many mapping. For each image, it only has one label. For each category, there may exist multiple hits.

5.	Some results are confusing. In Table 2, the variance of L2RCLIP in terms of Accuracy is the highest (7.2) while the MAE variance is relatively low (0.05).

6.	There are many inconsistent in the formulation and equations. For example, in line 147, M represents the class numbers and k is the length of rank templates. But in Eq 2, K is the class number and k is the class index. Again, in Line 182, k becomes the length of global context prompts.

7.	The equations from (3)-(6) are pretty messy. The sign in Eq 4 is wrong.

8.	Typos: Line 135, there is no Fig. 3.2.


**Questions:**

The key design has flaws. There are many typos and errors. This paper is highly unpolished.

---

> ### Author Rebuttal · Authors · 2023-08-09
>
> # Response to Reviewer VQC9
> We appreciate the reviewer's insightful comments. However, there seems to be some misunderstanding. We would like to clarify the issues and address the questions as follows.
>
> **Q1: The design of RankFormer.**
>
> **[Reply]** First, we want to clarify the misunderstanding regarding the training parameters in RankFormer. As described in Line 145-147, we **fix the parameters of rank prompts, with the $W$ being learnable while the $R$ is fixed**. Second, our ablation study demonstrates that our RankFormer outperforms the baseline when we make $R$ learnable parameters.
>
> **Q2: The total loss used during training.**
>
> **[Reply]** Due to page limitations, we have included this section in the supplementary materials. We utilize the cross-modal ordinal pairwise loss $L_{cop}$ and asymmetrical contrastive loss $L_{t2i}$ and $L_{i2t}$ to learn reliable rank prompts. In the second stage, we employ the cross-entropy loss $L_{ce}$ and simplified cross-modal ordinal pairwise loss $L_{scop}$ to fine-tune the image encoder. Further details can be found in Supp. Line 17-25.
>
> **Q3: Our setting for $L_{cop}$ and $L_{scop}$.**
>
> **[Reply]** When finetuning the image encoders, we fix the learned rank prompts so that the entropy estimation of these text embeddings can be ignored, which is equivalent to setting $\lambda=0$. It is worth noting that $L_{scop}$ is indeed a special case of $L_{cop}$, as you have suggested.
>
> **Q4: Many-to-many mapping.**
>
> **[Reply]** There seems to be a misunderstanding. We are employing pairwise contrastive learning, similar to CLIP. This means that the <category, image> and <image, category> pairs should have a similar many-to-many relationship from a row-wise or column-wise perspective. Thus, both the category and image may exist multiple hits.
>
> **Q5: Check the result on Adience dataset.**
>
> **[Reply]** We apologize for the oversight. We have rechecked the test pipeline for the Adience dataset, and the accurate results are as follows:
>
> |  L2RCLIP(Ours)   | Total  | 5-fold 01 | 5-fold 02 | 5-fold 03 | 5-fold 04 | 5-fold 05 |
> |  ----  | ----  | ----  | ----  | ----  | ----  | ----  |
> | MAE | 0.36±0.05 | 0.39 | 0.28 | 0.39 | 0.35 | 0.42 |
> | Accuracy  | 66.2%±4.4% | 64.8% | 73.5% | 64.6% | 68.0% | 60.1% |
>
> **Q6-8: The typos, formula and notation.**
>
> **[Reply]** We will ensure to avoid repeated notation and correct any mistakes in our revised paper.
>
> We have made every effort to address your comments faithfully in the revised paper. If you have any additional comments, please let us know. Thank you once again for your constructive feedback. We truly appreciate it.

---

> ### Author Response · Authors · 2023-08-18
> **Looking forward to the response from Reviewer VQC9**
>
> Dear Reviewer VQC9,
>
> We have tried our best to address all the concerns and provided as much evidence as possible. May we know if our rebuttals answer all your questions? We truly appreciate it.
>
> Best regards,
>
> Author #3203

---

> > ### Comment · Reviewer_VQC9 · 2023-08-19
> >
> > Q1: What do you mean R is fixed? So R is randomly initialized and then fixed for the whole pipeline? Or do you choose a two-stage framework to learn f_W(R) (one for R and one for W) even though this method already uses a two-stage framework to learn text and image separately?
> >
> > Q2: It's still unclear for readers. Why do the two stages have different losses? Why does stage 1 uses $L_{i2t}$ and  $L_{t2i}$ while stage 2 uses $L_{ce}$? How $\lambda $ is chosen for $L_{cop}$? There is no theoretical derivation or explanation here.
> >
> > One more question here. since $L_{cop}$ and $L_{scop}$ correspond to the loss functions of the two stages, respectively. So what's the setting in Table 5 when only one of these two losses is used?
> >
> > Many details about understanding methods and code implementation are missing or confusing, and there are errors and inconsistencies in the Equations, all of which give the impression that the submission was rushed through without careful polishing.

---

> > > ### Author Response · Authors · 2023-08-19
> > > **Response to Reviewer VQC9**
> > >
> > > Thanks for your reply. We would like to clarify the issues as follows.
> > > ***
> > > **Q1: Further explanation of R.**
> > >
> > > **[Reply]** In fact, R is initialized by tokenized rank templates (e.g. A photo of {age} years old face) and is fixed for the whole pipeline. The detailed initialization process is as follows. Suppose we have a rank template like “A photo of 21 years old face”. We first tokenize it into input ids by byte-level BPE, and then map the input ids into embedding vectors. **Then, we use these vectors to initialize the R and then fix it during the whole two-stage training.** We have also conducted the **initialization ablation study** in Table 6 of our manuscript to verify that our methods are robust to the diverse initialization strategy.
> > >
> > > **Q2: Further explanation of losses.**
> > >
> > > **[Reply]** **Reasons to use different losses.** In the first stage, text features exhibit semantic alignment with image features but lack satisfactory ordinal alignment with other text features. To enhance the ordering relationship of different text features, we employ RankFormer and the $L_{cor}$ loss. To preserve semantic alignment and prevent its destruction, we follow CLIP and use a variant of the contrastive loss, i.e., $L_{t2i}$ and $L_{i2t}$. In the second stage, as we observe promising ordinal alignment in the text features, we fix them and use our proposed $L_{scor}$ loss along with the standard classification loss, $L_{ce}$, to finetune the image encoder.
> > >
> > > **The connection of $L_{t2i} / L_{i2t}$ and $L_{ce}$.** As you have suggested, we agree that there is no obvious difference of $L_{t2i}/L_{i2t}$ and $L_{ce}$.  Both $L_{t2i}/L_{i2t}$ and $L_{ce}$ are likely to work in our settings. We default to using $L_{t2i}/L_{i2t}$ for text-image contrastive learning and $L_{ce}$ for the classification task.
> > >
> > > **The choice of $\lambda$.** As deduced in Eq 6, $\lambda$ should be set to $1$ in the $L_{cor}$. However, during the second-stage training, we freeze the text branch, causing the term in Eq 6 (i.e. $\lambda T_{y_i}^\top T_j$) to provide no gradient. Consequently, setting $\lambda=0$ is equivalent to this scenario. We will revise this part to enhance clarity in our revised version.
> > >
> > > **Q3: The ablation study for $L_{cor}$ and $L_{scor}$.**
> > >
> > > **[Reply]** We list all the experiment settings  in ablation study (Table 5 in our manuscript). All experiments follow a two-stage training approach. Rank prompts are initialized as described in Q1, while context prompts are initialized randomly. We denote $L_{t2i}/L_{i2t}$ as **I-A**, $L_{cor}$ as **I-B**, $L_{ce}$ as **II-A**, and $L_{scor}$ as **II-B**. The ablation study for $L_{cor}$ and $L_{scor}$ is highlighted in bold. We will provide ablation details in the revised version of our manuscript.
> > >
> > > **Table 1. The training setting in ablation study**
> > > |Ablation | Setting 0 | Setting 1 | Setting 2 | Setting 3 | Setting 4 | Setting 5 | Setting 6 | Setting 7 |
> > > |--- | --- | --- | --- | --- | --- | --- | --- | --- |
> > > |Rank prompts| Learnable | RankFormer | **Learnable** | **Learnable** | **Learnable** | **RankFormer** | **RankFormer** | **RankFormer** |
> > > |Context prompts| Learnable | Learnable | **Learnable** | **Learnable** | **Learnable** | **Learnable** | **Learnable** | **Learnable** |
> > > |Loss in Stage I| I-A  | I-A | **I-A,I-B** | **I-A** | **I-A,I-B** | **I-A,I-B** | **I-A** | **I-A,I-B** |
> > > |Loss in Stage II| II-A | II-A | **II-A** | **II-A,II-B** | **II-A,II-B** | **II-A** | **II-A,II-B** | **II-A,II-B** |
> > >
> > > We sincerely appreciate your response. We have made every effort to address your concerns, and we commit to including more details and code implementations in our revised version. Additionally, we will carefully address any typos and other errors.
> > >
> > > Please let us know if it addresses your concerns. Thank you again for your insightful comments. We truly appreciate them.

---

### Author Rebuttal · Authors · 2023-08-09

# Response to All Reviewers
Thank you for your valueble review and insightful suggestions. **The .pdf file includes extra figures. Please download it if needed.**
We have made every attempt to address your comments in the revised manuscript and hope that you find this revision satisfactory. If you have additional concerns, please let us know. We do appreciate them.

---

### Decision · Program_Chairs · 2023-09-21

**Decision:**

Accept (poster)

**Comment:**

Five out of six reviewers recommended the paper for acceptance (one of the positive reviews was mentioned in the discussion, but not updated in the system). The AC reviewed the paper, reviews, and rebuttals, and agrees with the consensus. The paper presents a novel ordering alignment method for ordinal classification leveraging strong priors derived from human language. The paper provides a thorough experimental validation, clearly outperforming prior state-of-the-art methods, and demonstrating the effectiveness of the proposed procedure.